# Federated Learning under Evolving Distribution Shifts

## Abstract

Federated learning (FL) is a distributed learning paradigm that facilitates training a global machine learning model without collecting the raw data from distributed clients. Recent advances in FL have addressed several considerations that are likely to transpire in realistic settings such as data distribution heterogeneity among clients. However, most of the existing works still consider clients' data distributions to be static or conforming to a simple dynamic, e.g., in participation rates of clients. In real FL applications, client data distributions change over time, and the dynamics, i.e., the evolving pattern, can be highly non-trivial. Further, evolution may take place from training to testing. In this paper, we address dynamics in client data distributions and aim to train FL systems from time-evolving clients that can generalize to future target data. Specifically, we propose two algorithms, *FedEvolve* and *FedEvp*, which are able to capture the evolving patterns of the clients during training and are test-robust under evolving distribution shifts. Through extensive experiments on both synthetic and real data, we show the proposed algorithms can significantly outperform the FL baselines across various network architectures.

## 1 Introduction

Federated learning (FL) is a widely used distributed learning framework where multiple clients, using their local data, train machine learning models collaboratively, orchestrated by a server (McMahan et al., 2017; Yang et al., 2019; Zhang et al., 2021a). A problem that has been extensively studied in FL literature is learning from heterogeneous clients, i.e., ensuring convergence of FL training and avoiding degradation of accuracy when clients' data are **not** identically and independently distributed (non i.i.d.) (Diao et al., 2021; Achituve et al., 2021; Reisizadeh et al., 2020).

Although a variety of approaches such as robust FL (Reisizadeh et al., 2020) and personalized FL (Wang et al., 2019) have been proposed to tackle the issue of data heterogeneity, most of them still assume that the data distribution of each client is *static* and, in particular, remains fixed between training and testing. Some recent works (Jiang & Lin, 2023; Gupta et al., 2022) move one step further by proposing test-robust FL models when there exist distribution shifts between training and testing data. However, they only consider *one-step* shift between training and testing while the training data distribution is still assumed to be static. In practice, FL systems are trained and deployed in dynamic environments that may continually change over time, e.g., satellite data evolve due to environmental changes, clinical data evolve due to changes in disease prevalence, etc. Existing FL algorithms without considering such evolving distribution shifts may result in inaccurate models and even fail to converge during the training phase.

In this paper, we will explore two questions:

- How can data stream with evolving distribution shifts impact FL systems (with or without client heterogeneity)?

- How can we exploit the evolving patterns from training data (source domains) and deploy our model on the unseen future distribution (target domain)?

The goal is to continuously train an FL model from distributed, time-evolving data that can generalize well on future target data. Figure 1 shows one motivating example.

Note that although the problem of learning under evolving distribution shifts has been studied recently in the centralized setting (typically known as evolving domain generalization), e.g., see Wang et al. (2022); Qin et al. (2022), it remains unclear how evolving distribution shifts can impact FL training and how to design FL algorithms when both evolving distribution shifts and data heterogeneity exist. The most relevant line of research to ours is continual federated learning (CFL) (Yoon et al., 2021; Casado et al., 2022), which aims to train an FL system continuously from a set of distributed time series. However, the primary objective of these works is to stabilize the training process and tackle the issue of catastrophic forgetting (i.e., prevent forgetting the previously learned old knowledge as the model is updated on new data). This differs from our work where we aim to explicitly learn evolving patterns and leverage them to adapt the model on future unseen data.

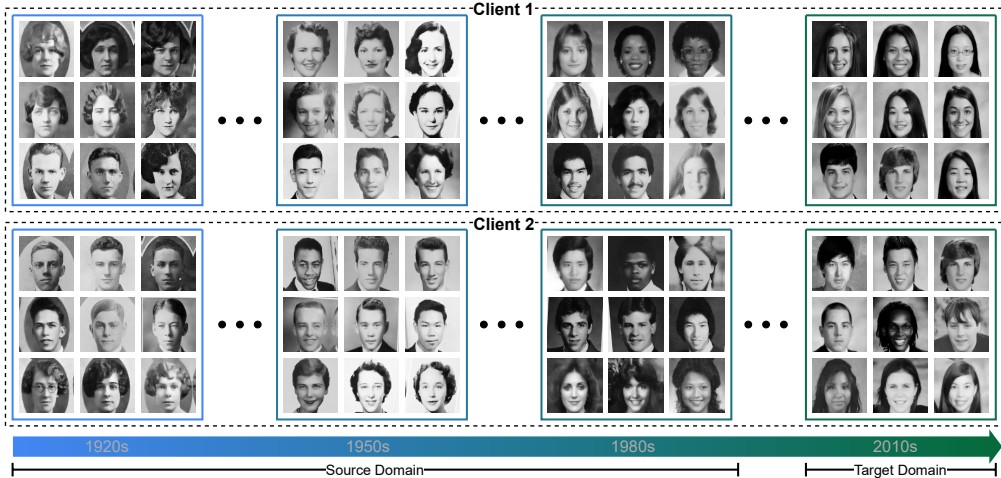

Figure 1: Illustration of evolving distribution shifts and client heterogeneity: Consider an FL system trained from distributed time-evolving photos (Ginosar et al., 2015) for gender classification. In this example, data exhibits obvious evolving patterns (e.g., changes in facial expression and hairstyle, improvement in the quality of images). Besides, clients are non-i.i.d and they have different class distributions. Our goal is to train an FL model that captures the evolving pattern of source domains and generalizes it to the future target domain.

To answer the above two questions, we will examine the performance of existing FL methods on time-evolving data, including a wide range of methods such as traditional FL methods, personalized FL methods, test-time adaptation methods, domain generalization methods, and continual FL methods. We observe that existing methods cannot capture evolving patterns and fail to generalize on future data. We then propose *FedEvolve*, an FL algorithm that learns the evolving patterns of clients during the training process and can generalize to future test data.

Specifically, *FedEvolve* learns the evolving pattern of source domains through *representation learning*. It assumes there exists a mapping function for each client that captures the transition of any two consecutive domains. To learn such transition, each client in *FedEvolve* learns two distinct representation mappings that map the inputs of domains in two consecutive time steps to a representation/latent space. By minimizing the distance between the distributions of these feature representations, *FedEvolve* captures the transition over two consecutive steps.

Although *FedEvolve* shows superior performance in learning from evolving distribution shifts in empirical experiments, the need for two distinct representation mappings brings double overhead during FL training. To reduce the computation cost and communication overhead, we further develop *FedEvp* as a more efficient and versatile version of *FedEvolve* by updating one representation mapping when evolving distribution shifts occur. Moreover, *FedEvp* better tackles heterogeneous data by incorporating the personalization strategy to partially personalize the model on each client's local data.

We illustrate via extensive experiments that our algorithms significantly outperform current benchmarks of FL when the feature domain is evolving, on multiple datasets (Rotated MNIST/EMNIST, Circle, Portraits, Caltran) using different models (MLP, CNN, ResNet). Our main contributions are:

- We identify the evolving distribution shift in FL that the current robust FL, personalized FL, and test-robust FL frameworks have failed to consider.
- We propose *FedEvolve* to actively capture the evolving pattern from evolving source domains and generalize to unseen target domains.
- We propose a more efficient and versatile version of algorithm *FedEvp* that learns domain-invariant representation from evolving prototypes.
- We empirically study how FL systems are affected when both evolving shifts and local heterogeneity exist. Experiments on multiple datasets show the superior performance of our methods compared to previous benchmark models.

## 2 Related Work

We briefly review related previous works in this section.

**Tackle client heterogeneity in FL.** Many approaches have been proposed to tackling data heterogeneity issues in FL and they can be roughly categorized into four classes. The first method is to add a regularization term. For example, Li et al. (2020; 2021b) proposed to steer the local models towards a global model by adding a regularization term to guarantee convergence when the data distributions among different clients are non-IID. The second method is clustering (Briggs et al., 2020; Ghosh et al., 2020; Sattler et al., 2020). By aggregating clients with similar distribution into the same cluster, the clients within the same cluster have lower statistical heterogeneity. Then, a cluster model that performs well for clients within this cluster can be found to reduce the performance degradation of statistical heterogeneity. The third method is to mix models or data. For example, Zhao et al. (2018) proposed a data-sharing mechanism where clients update models according to both the local train data and a small amount of globally shared data. Wu et al. (2022); Shin et al. (2020) developed mixup data augmentation techniques to let local devices decode the samples collected from other clients. Mansour et al. (2020) find a mixture of the local and global models according to a certain weight. The fourth method is robust FL. For instance, Reisizadeh et al. (2020); Deng et al. (2020b) obtain robust Federated learning models by finding the best model for worst-case performance. Notably, Reisizadeh et al. (2020) only considers the affine transformation of data distributions and Deng et al. (2020b) focuses on varying weight combinations over local clients. In addition, different personalization methods are applied to local clients, such as personalization (Wang et al., 2019; Yu et al., 2020; Arivazhagan et al., 2019; Huang et al., 2023; Bao et al., 2023), representation learning (Arivazhagan et al., 2019; Collins et al., 2021; Chen & Chao, 2022; Jiang & Lin, 2023), and meta-learning (Fallah et al., 2020).

**FL with dynamic data distributions.** While most previous works on statistical heterogeneity have considered static situations (i.e., the local heterogeneity stays constant during training), another line of literature focuses on FL in a dynamic environment where various distribution drifts occur. Some works aim to tackle drifts caused by time-varying participation rates of clients with local heterogeneity (Rizk et al., 2020; Park et al., 2021; Wang & Ji, 2022; Zhu et al., 2021), while other works assume the global distributions are also evolving (Guo et al., 2021; Casado et al., 2022; Yoon et al., 2021). However, among all previous works, Jiang & Lin (2023); Gupta et al. (2022) are the only ones considering the distribution shift between training and testing, but they assume the training distribution itself is static.

**Evolving domain generalization.** *Domain Generalization* (DG) has been extensively studied to generalize ML algorithms to unseen domains where different methods including data manipulation (Khirodkar et al., 2019; Robey et al., 2021), representation learning (Blanchard et al., 2017; Deshmukh et al., 2019), and domain adversarial learning (Rahman et al., 2020; Zhao et al., 2020). To go one step further, a few works have considered the evolving patterns of the domains (Hong Liu, 2020; Zhang & Davison, 2021; Kumar et al., 2020; Wang et al., 2022; Qin et al., 2022), but only Wang et al. (2022); Qin et al. (2022) consider *Evolving Domain Generalization* (EDG) where the target domain is not accessible. Wang et al. (2022) developed an algorithm to learn embeddings of the previous domain and the current domain such that their representations

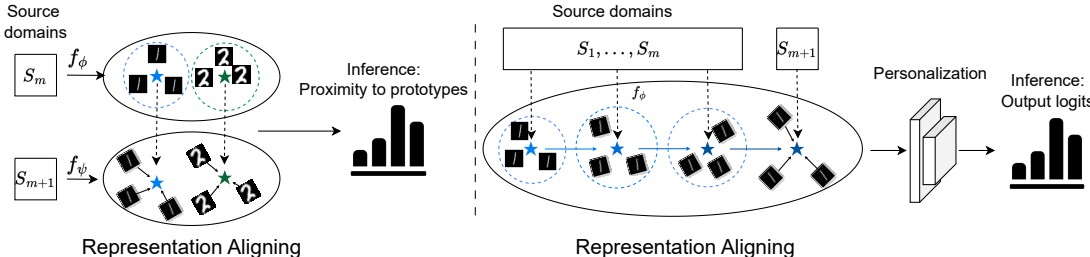

Figure 2: Illustration of *FedEvolve* (left) and *FedEvp* (right): (i). *FedEvolve* consists of two distinct modules $\phi$ and $\psi$, where $\phi$ calculates the prototypes for domain $S_m$, individually for each class, using mean values as class representations. Then, $\psi$ represents a data batch from the domain $S_{m+1}$. Both modules are updated based on the distance between $S_{m+1}$ representations and $S_m$ prototypes. During inference, $\psi$ computes the distance to the latest domain's prototypes, then selects the minimal one as the prediction result. (ii). *FedEvp* simplifies *FedEvolve* by removing $\psi$ and integrating a classifier $w$ with $\phi$. This decreases the communication cost during federated training. Instead of using localized prototypes from just $S_m$, *FedEvp* builds global prototypes from domains $S_1$ to $S_m$. These prototypes align with the representations of the succeeding domain $S_{m+1}$, providing an integrated feature representation across diverse domains. By emphasizing consistent feature representation, *FedEvp* ensures its classifier adeptly handles an unseen domain, making predictions resilient and versatile across changing data contexts.

are invariant. Qin et al. (2022) developed a dynamic probabilistic framework to model the underlying latent variables across domains. Wang et al. (2020) considers a similar problem under the domain adaptation setting, where they use domain discriminators to learn domain-invariant features and adapt the model to target domains. However, all these previous works consider the *centralized setting*. Thus, there is a gap for EDG under distributed settings, and in particular for FL.

## 3 Problem Formulation

Consider a federated learning (FL) system consisting of $K$ clients, whose data distributions vary dynamically over time. Define $\{S_1, ..., S_M\}$ as $M$ consecutive domains that characterize the evolution of the clients' global distribution. Let $\mathcal{D}_m^k$ be the local dataset of client $k \in \{1, ..., K\}$ at $m$-th domain. The clients are heterogeneous and they may have access to different class labels. Given an FL model with parameter $h$, let $\ell(x, y; h)$ be the corresponding loss evaluated on a labeled data sample $(x, y)$. Our goal is to learn an FL model $h$ from $K$ clients over $M$ domains that can generalize on a subsequent target domain $S_{M+1}$. That is, we wish to find $h^*$ that minimizes the total loss at target domain $S_{M+1}$ over $K$ clients:

$$h^* = \underset{h \in \mathbb{R}^d}{\arg\min} \sum_{k=1}^{K} \alpha_k L_k(h) \tag{1}$$

where $\alpha_k$ is the weight of client $k$ (e.g., the proportion of sample size), and $L_k(h) := \mathbb{E}_{(x_i, y_i) \sim \mathcal{D}_{M+1}^k} \ell(x_i, y_i; h)$ is the local loss of client $k$ evaluated on target domain $S_{M+1}$.

## 4 Methodology

To learn an FL model from time-evolving data that generalizes well to the future domain, we need to learn the evolving pattern of source domains during federated training. Motivated by (Wang et al., 2022; Snell et al., 2017), we assume there is a pattern in the transition between every two consecutive domains $S_m$ and $S_{m+1}$. Instead of learning evolving patterns directly in the input space, we consider *representation learning* to learn the evolution in a representation space. Next, we introduce two algorithms *FedEvolve* and *FedEvp*, which align data representation from evolving domains and facilitate local personalization. Specifically, *FedEvolve* is designed to actively identify the evolving pattern between two consecutive domains, while *FedEvp* first learns invariant representation across all existing domains, then generalizes to the unknown evolving domain.

### 4.1 FedEvolve

To actively capture the evolving patterns of source domains, *FedEvolve* learns two distinct learnable representation functions $f_\phi, f_\psi$. Given two consecutive domain $S_m$ and $S_{m+1}$:

- $f_\phi(S_m)$ is the **estimated representation** of **subsequent** domain $S_{m+1}$ using input $S_m$.

- $f_\psi(S_{m+1})$ is the **representation** of input domain $S_{m+1}$.

Because $f_\phi$ estimates the representation of a domain using the previous domain, , we can use it to estimate unknown target domain $S_{M+1}$ from source domains $\{S_1, ..., S_M\}$. Let $\phi, \psi$ be the trainable neural network parameters of $f_\phi, f_\psi$, respectively. To learn the evolving pattern, we aim to learn $\phi, \psi$ such that the estimated future domain representation $f_\phi(S_m)$ is sufficiently accurate and close to the actual representation $f_\psi(S_{m+1})$, i.e., we need minimize the distance between $f_\phi(S_m)$ and $f_\psi(S_{m+1})$. Inspired by (Wang et al., 2022), to align the two representations while capturing the class characteristics across evolving domains, we leverage prototypical learning (Snell et al., 2017) to directly align their representation prototypes.

Specifically, for each client $k$ and domain $S_m$, we define the *prototype* $C_{m,y}^k$ of class $y$ on the client's local dataset $\mathcal{D}_m^k$ as the mean value of the representations produced by $f_{\widetilde{\phi}_k}$, where $\widetilde{\phi}_k$ is the local parameter learned on client $k$, i.e.,

$$C_{m,y}^k = \frac{1}{|\mathcal{D}_{m,y}^k|} \sum_{x \in \mathcal{D}_{m,y}^k} f_{\widetilde{\phi}_k}(x) \tag{2}$$

where $\mathcal{D}_{m,y}^k \subseteq \mathcal{D}_m^k$ is a subset of data instances with label $y$, $|\mathcal{D}_{m,y}^k|$ is the cardinality of this set. For the next domain $S_{m+1}$, *FedEvolve* minimizes the distance between its representation $f_{\widetilde{\psi}_k}(S_{m+1})$ and $f_{\widetilde{\phi}_k}(S_m)$ estimated from $S_m$. This can be achieved by aligning the representation $f_{\widetilde{\psi}_k}$ for data points from the domain $S_{m+1}$ to its corresponding class prototype $C_{m,y}^k$. Mathematically, we minimize the loss defined below:

$$\ell(x, y) = \log \frac{\exp\left(-d\left(f_{\widetilde{\psi}_k}(x), C_{m,y}^k\right)\right)}{\sum_{y' \in \mathcal{Y}_{\mathcal{D}_{m+1}^k}} \exp\left(-d\left(f_{\widetilde{\psi}_k}(x), C_{m,y'}^k\right)\right)} \tag{3}$$

where $(x, y)$ is a sample pair from $\mathcal{D}_{m+1}^k$ and $\mathcal{Y}_{\mathcal{D}_{m+1}^k}$ including all class labels in $\mathcal{D}_{m+1}^k$. $d$ is a distance measure (e.g. Euclidean distance, cosine distance) that quantifies the difference between the feature representation $f_{\widetilde{\psi}_k}(x)$ and the prototype $C_{m,y}^k$ of class $y$ from the local dataset $\mathcal{D}_m^k$. In this paper, we employ Euclidean distance.

By minimizing Eqn. equation 3 on all active clients, local models learn the evolving pattern by aligning representations of domain $S_{m+1}$ with prototypes from the former domain $S_m$. After local updates, active clients $\mathcal{I}_t$ send local parameters to the server and the server performs an average aggregation to update the global parameters $\phi = \frac{1}{|\mathcal{I}_t|} \sum_{k \in \mathcal{I}_t} \widetilde{\phi}_k, \psi = \frac{1}{|\mathcal{I}_t|} \sum_{k \in \mathcal{I}_t} \widetilde{\psi}_k$. These aggregations encapsulate global information with diverse data contributions of all clients. Once consolidated, these models can be directly dispatched to the clients and facilitate continuous model adaptations to the evolving data distributions across the federated network.

After training on source domains, we can use the learned representation functions $f_\phi, f_\psi$ to predict the target domain $S_{M+1}$. Specifically, we first compute the prototypes of $f_\phi(S_M)$ on $S_M$. Then, we apply $f_\psi$ to test samples in $S_{M+1}$ to generate representations $f_\psi(S_{M+1})$ and classify them based on proximity to prototypes. We present the pseudocode of *FedEvolve* in Algorithm 1 in Appendix A.

### 4.2 FedEvp

Because the two distinct representation functions $f_\phi$ and $f_\psi$ in *FedEvolve* are usually large neural networks such as ResNet (He et al., 2016) in real-world image datasets, there is a non-negligible additional overhead to transmit extra parameters of the second representation mapping, rendering deployment challenges in environments with limited computational resources or network bandwidth. To address the potential overhead, we also present *FedEvp*, an efficient and streamlined strategy that achieves similar performance as *FedEvolve*.

Unlike the dual model mechanism of *FedEvolve*, *FedEvp* adopts a single-model strategy to reduce communication costs while simultaneously accelerating training. As shown in the right plot of Figure 2, *FedEvp* aims learn the evolving-domain-invariant representation using a representation function $f_\phi$ by continuously aligning data to prototypes from previous domains. If we can develop a representation that is resilient to evolving distributional shifts, a single classifier could effectively serve all domains. To further address local heterogeneity, we also incorporate an efficient personalization step for the classifier.

To ensure a consistent learning process, *FedEvp* maintains evolving prototypes according to the classes of consecutive domains. In essence, the prototypes learned by *FedEvp* consolidate the global information from all previous domains to enable the learning of domain-invariant features. For each class $y$ within client $k$, an evolving prototype $C_{m,y}^k$ is established as follows: $C_{0,y}^k$ is set to zero; for other domains ranging from 1 to $M$, the prototype is updated as Eqn. equation 4,

$$C_{m,y}^k = \frac{(m-1)}{m} C_{m-1,y}^k + \frac{1}{m} \left( \frac{1}{|\mathcal{D}_{m,y}^k|} \right) \sum_{x \in \mathcal{D}_{m,y}^k} f_{\widetilde{\phi}_k}(x) \tag{4}$$

where $\mathcal{D}_{m,y}^k$ is the set of all instances in the current domain $m$ that belongs to class $y$, and $f_{\widetilde{\phi}_k}(x_i)$ denotes the representation of instance $x_i$ under the client $k$'s local model parameters $\widetilde{\phi}_k$. Such an iterative update mechanism ensures that the prototype $C_{m,y}^k$ evolves as new domains are introduced, gradually incorporating information from each one. As a result, $C_{M,y}^k$ becomes a representative prototype of class $y$ across all available training domains for client $k$.

We then align the data from domain $S_{m+1}$ to the prototypes $C_m^k$ to update parameter $\phi$. We adopt the same loss function as *FedEvolve* given in Eqn. equation 5,

$$\ell_f(x,y) = \log \frac{\exp\left(-d\left(f_{\widetilde{\phi}_k}(x), C_{m,y}^k\right)\right)}{\sum_{y' \in \mathcal{Y}_{\mathcal{D}_{m+1}^k}} \exp\left(-d\left(f_{\widetilde{\phi}_k}(x), C_{m,y'}^k\right)\right)} \tag{5}$$

where $d$ is the same distance metric as in *FedEvolve*. And $d(f_{\widetilde{\phi}_k}(x), C_{m,y}^k)$ is the distance between the feature representation $f_{\widetilde{\phi}_k}(x)$ of instance $x$ and the prototype $C_{m,y}^k$ of class $y$, $\mathcal{Y}_{\mathcal{D}_{m+1}^k}$ is the set of classes in the $m+1$ domain.

Besides minimizing $\ell_f$ to learn domain-invariant representation, we introduce a classifier $\widetilde{w}_k$ which is updated by minimizing empirical risk $\ell_e$ defined as:

$$\ell_e(x,y) = -y \log \frac{\exp\left(g_{\widetilde{w}_k}^y\left(f_{\widetilde{\phi}_k}(x)\right)\right)}{\sum_{y' \in \mathcal{Y}_{\mathcal{D}_m^k}} \exp\left(g_{\widetilde{w}_k}^{y'}\left(f_{\widetilde{\phi}_k}(x)\right)\right)} \tag{6}$$

where $g_{\widetilde{w}_k}^y\left(f_{\widetilde{\phi}_k}(x)\right)$ is the predicted outputs of the class $y$ for instance $(x,y) \in \mathcal{D}_{m,y}^k$, computed by the classifier $\widetilde{w}_k$. In our experiments, $\ell_e$ is the classical cross-entropy loss.

After local updates, *FedEvp* aggregates the local parameters at the server $\phi = \frac{1}{|\mathcal{I}_t|} \sum_{k \in \mathcal{I}_t} \widetilde{\phi}_k, w = \frac{1}{|\mathcal{I}_t|} \sum_{k \in \mathcal{I}_t} \widetilde{w}_k$. These aggregated global models are then sent back to clients for future updates. As *FedEvp* relies on the classifier using evolving domain invariant features instead of directly using the difference between two consecutive domain representations, the prediction may be influenced by the client's heterogeneity. To handle the issue raised by local heterogeneity, a personalization mechanism, akin to *local fine-tuning*, is further incorporated. Specifically, we personalize each client by updating both the classifier $\widetilde{w}$ and the *last layer* of the feature extractor $\widetilde{f_\phi}$ for an additional epoch on the client's local dataset. The pseudocode of *FedEvp* is given in Algorithm 2 in Appendix A.

## 5   Experiments

To evaluate our methods, we consider classification tasks using various network architectures and report average accuracy and standard deviation over three runs. The detailed implementation can be found in Appendix C. Dirichlet distribution is used to control the level of heterogeneity with parameter $Dir \in [0, \infty)$. The smaller Dir implies that the clients are more heterogeneous. Heterogeneous clients may have access to different class labels. We report the average performance across clients and the performance on the server. Both are evaluated on the test domain after the last epoch. The federated training phase follows typical FL steps. In each communication round $t$, a subset $\mathcal{I}_t$ of $K$ clients join the system and the server distributes aggeragated global model parameters to client $k \in \mathcal{I}_t$. Upon receiving these parameters, each client $k$ initializes its local parameters to those and performs $\tau$ local updates.

### 5.1   Datasets and Networks

We evaluate *FedEvolve* and *FedEvp* on both **synthetic data** (Circle) and **real data** (Rotated MNIST, Rotated EMNIST, Portraits, and Caltran). All datasets either come with evolving patterns or are adapted to evolving environments. For all datasets, the last domain is viewed as the target domain. The feature extractor in the neural network is viewed as $\phi$ and $\psi$, and the classifier is $w$ mentioned in the previous section.

**Circle (Pesaranghader & Viktor, 2016).** This synthetic data has 30 evolving domains. 30000 instances within these domains are sampled from 30 two-dimensional Gaussian distributions, with the same variance but different means that are uniformly distributed on a half-circle. We use a 5-layer multilayer perception (MLP) with 3 layers serving as a representation function ($f_\phi$ and $f_\psi$ in FedEvolve, $f_\phi$ in FedEvp) and the remaining 2 layers as a classifier ($f_w$ in FedEvp).

**Rotated MNIST (Ghifary et al., 2015) and Rotated EMNIST (Cohen et al., 2017).** The Rotated MNIST is a variation of the MNIST data, where we rotate the original handwritten digit images to produce different domains. Specifically, we partition the data into 12 domains and rotate the images within each domain by an angle $\theta$, beginning at $\theta = 0°$ and progressing in 15-degree increments up to $\theta = 165°$. We also consider other increments spanning from $0°$ to $25°$ to simulate varying degrees of evolving shifts. EMNIST is a more challenging alternative to MNIST with more classes including both hand-written digits and letters. We use the hand-written letters subset and split it into 12 domains by rotating images with a degree of $\theta = \{0°, 8°, ..., 88°\}$. We design a model consisting of a 4-layer convolutional neural network (CNN) for representation layers, followed by two linear layers for classification.

**Portraits (Ginosar et al., 2015).** It is a real dataset consisting of frontal-facing American high school yearbook photos over a century. This time-evolving dataset reflects the changes in fashion (e.g., high style and smile). We resize images to $32 \times 32$ and split the dataset by every 12 years into 9 domains. We use WideResNet (Zagoruyko & Komodakis, 2016) as the representation function to train the gender classifier. Note that the data is only intended to compare various methods.

**Caltran (Hoffman et al., 2014).** This real surveillance dataset comprises images captured by a fixed traffic camera. We divide the dataset into 12 domains where the samples from every 2-hour block form a domain (evolving shifts arising from changes in light intensity). ResNet18 (He et al., 2016) backbone is used as the representation function and the last linear layer is used as the classifier.

### 5.2   Baselines

We compare *FedEvolve* and *FedEvp* with various existing FL methods. These baselines cover a broad range of methods including traditional FL, strong personalized FL (PFL), centralized test-time adaptation (TTA) methods, federated TTA methods, and continual federated learning methods.

- *FedAvg* (McMahan et al., 2017): A FL method that learns the global model by averaging the client's local model.

- *GMA* (Tenison et al., 2022): A FL method using gradient masked averaging approach to aggregate local models.

Table 1: Average accuracy over three runs of experiments on rotated MNIST under i.i.d and non-i.i.d distribution. The client heterogeneity(Dir) is determined by the value of Dirichlet distribution (Yurochkin et al., 2019).

| Method | Dir$\to \infty$ | | Dir=1.0 | | Dir=0.1 | |
|---|---|---|---|---|---|---|
| | Client | Server | Client | Server | Client | Server |
| FedAvg | $65.92_{\pm1.01}$ | $66.34_{\pm0.34}$ | $62.35_{\pm0.97}$ | $63.16_{\pm1.78}$ | $51.68_{\pm0.73}$ | $51.59_{\pm2.48}$ |
| GMA | $65.94_{\pm0.91}$ | $66.17_{\pm0.21}$ | $61.49_{\pm0.30}$ | $61.68_{\pm0.66}$ | $50.86_{\pm1.15}$ | $51.32_{\pm2.47}$ |
| Memo(G) | $65.94_{\pm1.34}$ | $66.78_{\pm2.30}$ | $61.39_{\pm0.94}$ | $62.91_{\pm2.55}$ | $49.76_{\pm5.58}$ | $52.06_{\pm1.23}$ |
| FedAvgFT | $48.70_{\pm1.03}$ | $66.61_{\pm0.59}$ | $57.95_{\pm2.91}$ | $62.61_{\pm1.02}$ | $69.51_{\pm1.97}$ | $51.59_{\pm1.70}$ |
| APFL | $62.37_{\pm1.08}$ | $65.57_{\pm1.54}$ | $67.58_{\pm1.09}$ | $63.98_{\pm2.31}$ | $70.37_{\pm2.19}$ | $50.66_{\pm0.47}$ |
| FedRep | $60.04_{\pm1.00}$ | $68.09_{\pm3.10}$ | $63.95_{\pm0.75}$ | $63.49_{\pm2.62}$ | $76.35_{\pm1.67}$ | $52.25_{\pm1.75}$ |
| Ditto | $65.23_{\pm0.87}$ | $65.35_{\pm1.50}$ | $68.14_{\pm0.92}$ | $64.64_{\pm1.45}$ | $75.55_{\pm2.56}$ | $50.89_{\pm2.79}$ |
| FedRod | $52.30_{\pm1.87}$ | $67.93_{\pm1.05}$ | $54.00_{\pm3.98}$ | $63.32_{\pm2.33}$ | $64.11_{\pm3.68}$ | $53.02_{\pm1.22}$ |
| Memo(P) | $51.70_{\pm2.48}$ | $65.35_{\pm1.47}$ | $59.84_{\pm0.61}$ | $64.75_{\pm1.59}$ | $69.46_{\pm2.77}$ | $50.27_{\pm2.85}$ |
| T3A | $53.94_{\pm0.76}$ | $66.61_{\pm0.59}$ | $61.60_{\pm2.49}$ | $62.61_{\pm1.02}$ | $71.73_{\pm1.63}$ | $51.59_{\pm1.70}$ |
| FedTHE | $66.84_{\pm1.51}$ | $67.43_{\pm0.23}$ | $67.98_{\pm0.43}$ | $62.55_{\pm1.98}$ | $78.52_{\pm3.92}$ | $53.40_{\pm0.74}$ |
| Flute | $62.97_{\pm1.39}$ | $63.27_{\pm1.13}$ | $68.86_{\pm0.75}$ | $61.46_{\pm0.16}$ | $78.44_{\pm3.54}$ | $54.71_{\pm3.28}$ |
| FedSR | $69.91_{\pm1.14}$ | $71.79_{\pm1.75}$ | $67.00_{\pm1.23}$ | $68.01_{\pm2.65}$ | $61.49_{\pm2.60}$ | $59.88_{\pm3.54}$ |
| CFL | $63.75_{\pm0.98}$ | $64.33_{\pm2.17}$ | $60.29_{\pm1.85}$ | $60.82_{\pm1.97}$ | $50.76_{\pm1.41}$ | $51.04_{\pm2.49}$ |
| CFeD | $70.22_{\pm0.63}$ | $71.66_{\pm0.66}$ | $68.07_{\pm0.72}$ | $68.64_{\pm1.38}$ | $60.41_{\pm2.33}$ | $61.27_{\pm2.93}$ |
| **FedEvolve** | $\mathbf{84.75}_{\pm1.39}$ | $\mathbf{84.43}_{\pm1.21}$ | $\mathbf{79.93}_{\pm1.00}$ | $\mathbf{77.25}_{\pm1.82}$ | $\mathbf{83.86}_{\pm1.81}$ | $\mathbf{71.66}_{\pm1.95}$ |
| **FedEvp** | $\mathbf{75.99}_{\pm0.31}$ | $\mathbf{77.63}_{\pm1.99}$ | $\mathbf{77.91}_{\pm1.80}$ | $\mathbf{73.85}_{\pm1.53}$ | $\mathbf{83.15}_{\pm0.49}$ | $\mathbf{61.84}_{\pm3.34}$ |

- *FedAvg + FT*: Fine-tunes the global model on local training data, an effective strategy for personalization in FL.

- *MEMO* (Zhang et al., 2022): A TTA method and we adapt it to FL. Following (Jiang & Lin, 2023), we term MEMO applied to the global model as MEMO(G) and to the FedAvg + FT model as MEMO(P).

- *APFL* (Deng et al., 2020a): A PFL method that leverages a weighted ensemble of personalized and global models.

- *FedRep* (Collins et al., 2021) and *FedRoD* (Chen & Chao, 2021): PFL methods that use a decoupled feature extractor and classifier to enhance personalization in FL.

- *Ditto* (Li et al., 2021a): A fairness-aware PFL method that has been shown to outperform other fairness FL methods.

- *T3A* (Iwasawa & Matsuo, 2021): A TTA method that is adapted to personalized FL by adding test-time adaptation to *FedAvg + FT*.

- *FedTHE* (Jiang & Lin, 2023): A TTA PFL method that tackles the data heterogeneity issue while learning test-time robust FL under distribution shifts.

- *FedSR* (Nguyen et al., 2022): A TTA FL method using the regular domain generalization method.

- *CFL* (Guo et al., 2021): A continual federated learning method that learns from time-series data while preventing catastrophic forgetting.

- *CFeD* (Ma et al., 2022): It uses distillation to tackle catastrophic forgetting in continual federated learning.

- *Flute* (Liu et al., 2024): Flute is a PFL method that facilitates the distillation of the subspace spanned by the global optimal representation from the misaligned local representations.

### 5.3 Results

In Figure 3, we examine how the algorithm performance changes as the degree of evolving shifts varies. Tables 1, 2 and 3 show the comparison with baselines, where we report both the averaged performance of clients' local models and the performance of the global model at the server. We also extend the experiments in Table 3 to the setting when clients are heterogeneous (Dir = 1.0) and present the results in Table 4.

**Impacts of distribution shifts and local heterogeneity.** First, we examine the impact of distribution shifts and client heterogeneity on FL systems. Figure 3 presents the results on RMNIST data under clients

Table 2: Average accuracy over three runs of experiments on rotated EMNIST-Letter under i.i.d and non-i.i.d distribution.

| | Dir→ $\infty$ | | Dir=1.0 | | Dir=0.1 | |
|---|---|---|---|---|---|---|
| Method | Client | Server | Client | Server | Client | Server |
| FedAvg | $53.83_{\pm1.84}$ | $54.18_{\pm1.72}$ | $52.72_{\pm4.45}$ | $52.77_{\pm3.74}$ | $46.72_{\pm2.55}$ | $45.71_{\pm1.77}$ |
| GMA | $54.23_{\pm1.77}$ | $55.10_{\pm1.71}$ | $51.23_{\pm1.93}$ | $51.42_{\pm0.79}$ | $48.40_{\pm1.75}$ | $48.61_{\pm2.13}$ |
| Memo(G) | $53.32_{\pm1.38}$ | $53.85_{\pm0.72}$ | $50.33_{\pm2.06}$ | $50.37_{\pm1.10}$ | $47.53_{\pm2.09}$ | $47.20_{\pm1.86}$ |
| FedAvgFT | $44.20_{\pm2.54}$ | $54.09_{\pm1.30}$ | $52.16_{\pm4.62}$ | $53.82_{\pm2.13}$ | $66.96_{\pm0.68}$ | $46.87_{\pm0.60}$ |
| APFL | $44.98_{\pm1.57}$ | $54.33_{\pm1.12}$ | $49.84_{\pm1.48}$ | $50.99_{\pm0.62}$ | $66.80_{\pm0.37}$ | $46.42_{\pm2.58}$ |
| FedRep | $39.01_{\pm2.03}$ | $46.39_{\pm2.49}$ | $47.26_{\pm2.64}$ | $47.25_{\pm0.93}$ | $67.51_{\pm1.35}$ | $44.12_{\pm0.46}$ |
| Ditto | $42.38_{\pm1.77}$ | $53.90_{\pm1.20}$ | $53.80_{\pm1.89}$ | $56.22_{\pm1.58}$ | $72.66_{\pm0.61}$ | $55.48_{\pm1.94}$ |
| FedRod | $44.25_{\pm1.60}$ | $51.79_{\pm2.77}$ | $49.53_{\pm0.81}$ | $50.32_{\pm2.61}$ | $67.31_{\pm2.03}$ | $45.74_{\pm3.99}$ |
| Memo(P) | $45.42_{\pm2.39}$ | $53.47_{\pm1.33}$ | $51.23_{\pm4.94}$ | $51.10_{\pm1.10}$ | $68.37_{\pm1.48}$ | $47.73_{\pm2.26}$ |
| T3A | $48.80_{\pm2.84}$ | $54.49_{\pm0.46}$ | $55.93_{\pm2.28}$ | $53.29_{\pm1.12}$ | $71.80_{\pm1.95}$ | $52.08_{\pm2.84}$ |
| FedTHE | $52.40_{\pm3.87}$ | $53.27_{\pm3.60}$ | $58.08_{\pm1.44}$ | $53.45_{\pm1.87}$ | $69.34_{\pm2.10}$ | $46.15_{\pm2.17}$ |
| Flute | $48.89_{\pm3.04}$ | $51.52_{\pm4.63}$ | $55.10_{\pm3.88}$ | $46.71_{\pm2.73}$ | $64.99_{\pm3.35}$ | $40.27_{\pm3.01}$ |
| FedSR | $55.71_{\pm0.09}$ | $56.92_{\pm0.44}$ | $51.40_{\pm4.65}$ | $55.35_{\pm3.93}$ | $44.38_{\pm2.30}$ | $49.43_{\pm2.48}$ |
| CFL | $40.65_{\pm2.19}$ | $41.41_{\pm1.86}$ | $45.82_{\pm2.34}$ | $46.13_{\pm1.01}$ | $40.24_{\pm3.50}$ | $39.37_{\pm4.29}$ |
| CFeD | $56.76_{\pm0.65}$ | $56.17_{\pm1.39}$ | $55.50_{\pm4.33}$ | $55.53_{\pm5.73}$ | $47.20_{\pm1.37}$ | $47.76_{\pm2.22}$ |
| **FedEvolve** | $\mathbf{83.58}_{\pm1.45}$ | $\mathbf{82.91}_{\pm1.36}$ | $\mathbf{82.13}_{\pm0.48}$ | $\mathbf{78.68}_{\pm0.25}$ | $\mathbf{87.67}_{\pm0.55}$ | $\mathbf{72.85}_{\pm1.03}$ |
| **FedEvp** | $\mathbf{67.30}_{\pm1.35}$ | $\mathbf{71.94}_{\pm1.50}$ | $\mathbf{73.61}_{\pm1.70}$ | $\mathbf{68.91}_{\pm0.30}$ | $\mathbf{87.01}_{\pm0.22}$ | $\mathbf{58.73}_{\pm0.96}$ |

Table 3: Average accuracy across various datasets over three runs. We consider the i.i.d setting that Dir→ $\infty$.

| | Circle | | Portraits | | Caltran | |
|---|---|---|---|---|---|---|
| | Client | Server | Client | Server | Client | Server |
| FedAvg | $70.40_{\pm6.51}$ | $70.40_{\pm6.51}$ | $94.10_{\pm0.13}$ | $94.10_{\pm0.13}$ | $62.93_{\pm2.10}$ | $64.31_{\pm2.13}$ |
| GMA | $62.55_{\pm6.94}$ | $62.55_{\pm6.94}$ | $94.18_{\pm0.14}$ | $94.18_{\pm0.14}$ | $63.28_{\pm3.48}$ | $63.85_{\pm3.49}$ |
| Memo(G) | - | - | $94.38_{\pm0.07}$ | $94.63_{\pm0.31}$ | $63.41_{\pm2.81}$ | $63.82_{\pm2.92}$ |
| FedAvgFT | $60.85_{\pm3.07}$ | $63.55_{\pm5.67}$ | $90.99_{\pm0.74}$ | $93.21_{\pm1.86}$ | $63.82_{\pm0.70}$ | $63.98_{\pm3.22}$ |
| APFL | $59.90_{\pm2.48}$ | $63.55_{\pm5.67}$ | $90.54_{\pm0.29}$ | $94.64_{\pm0.16}$ | $62.11_{\pm1.85}$ | $63.17_{\pm3.29}$ |
| FedRep | $64.37_{\pm5.60}$ | $64.97_{\pm6.05}$ | $90.88_{\pm0.63}$ | $93.50_{\pm1.15}$ | $62.03_{\pm3.05}$ | $64.07_{\pm2.41}$ |
| Ditto | $62.60_{\pm2.64}$ | $63.10_{\pm6.00}$ | $91.46_{\pm0.13}$ | $94.07_{\pm0.30}$ | $62.44_{\pm2.59}$ | $63.58_{\pm3.43}$ |
| FedRod | $64.60_{\pm2.33}$ | $65.00_{\pm6.55}$ | $91.57_{\pm0.18}$ | $94.78_{\pm0.43}$ | $64.14_{\pm3.94}$ | $58.29_{\pm4.75}$ |
| Memo(P) | - | - | $91.30_{\pm0.16}$ | $94.34_{\pm0.28}$ | $63.66_{\pm2.93}$ | $63.58_{\pm3.43}$ |
| T3A | $62.20_{\pm4.11}$ | $66.50_{\pm4.95}$ | $91.84_{\pm0.61}$ | $94.59_{\pm0.34}$ | $63.90_{\pm0.60}$ | $63.98_{\pm3.22}$ |
| FedTHE | $64.03_{\pm4.79}$ | $63.27_{\pm5.05}$ | $94.13_{\pm0.24}$ | $93.48_{\pm0.98}$ | $60.48_{\pm1.44}$ | $58.17_{\pm3.18}$ |
| Flute | $65.69_{\pm3.81}$ | $63.04_{\pm3.31}$ | $94.25_{\pm0.10}$ | $94.53_{\pm0.18}$ | $61.71_{\pm2.19}$ | $61.46_{\pm3.70}$ |
| FedSR | $72.77_{\pm3.38}$ | $71.62_{\pm5.70}$ | $94.43_{\pm0.35}$ | $94.52_{\pm0.35}$ | $64.57_{\pm1.36}$ | $66.02_{\pm1.47}$ |
| CFL | $72.12_{\pm8.76}$ | $72.12_{\pm8.76}$ | $92.91_{\pm1.07}$ | $92.91_{\pm1.07}$ | $63.68_{\pm3.61}$ | $63.92_{\pm3.15}$ |
| CFeD | $71.60_{\pm6.77}$ | $71.60_{\pm6.77}$ | $93.64_{\pm0.27}$ | $93.64_{\pm0.27}$ | $63.48_{\pm3.87}$ | $63.55_{\pm3.27}$ |
| **FedEvolve** | $\mathbf{84.25}_{\pm2.45}$ | $\mathbf{81.64}_{\pm1.95}$ | $\mathbf{95.43}_{\pm0.17}$ | $\mathbf{96.88}_{\pm1.35}$ | $\mathbf{65.04}_{\pm1.66}$ | $63.54_{\pm0.74}$ |
| **FedEvp** | $\mathbf{73.30}_{\pm5.02}$ | $\mathbf{74.12}_{\pm6.93}$ | $93.54_{\pm0.19}$ | $\mathbf{94.92}_{\pm0.11}$ | $\mathbf{66.59}_{\pm1.44}$ | $\mathbf{66.34}_{\pm0.69}$ |

Table 4: Accuracy of baselines across various datasets over three runs (Dir=1.0).

| | Circle | | Portraits | | Caltran | |
| --- | --- | --- | --- | --- | --- | --- |
| | Client | Server | Client | Server | Client | Server |
| FedAvg | $66.53_{\pm4.74}$ | $66.53_{\pm4.74}$ | $94.37_{\pm0.86}$ | $94.37_{\pm0.86}$ | $66.34_{\pm2.41}$ | $65.12_{\pm4.87}$ |
| GMA | $65.93_{\pm6.01}$ | $65.93_{\pm6.01}$ | $93.75_{\pm0.68}$ | $93.75_{\pm0.68}$ | $65.12_{\pm1.95}$ | $63.05_{\pm4.51}$ |
| Memo(G) | - | - | $93.81_{\pm0.45}$ | $93.81_{\pm0.45}$ | $66.20_{\pm2.07}$ | $65.54_{\pm0.73}$ |
| FedAvgFT | $65.97_{\pm1.49}$ | $66.93_{\pm3.30}$ | $92.54_{\pm0.65}$ | $94.56_{\pm0.43}$ | $65.12_{\pm2.84}$ | $65.56_{\pm3.61}$ |
| APFL | $64.23_{\pm0.80}$ | $66.93_{\pm3.30}$ | $92.16_{\pm0.42}$ | $94.47_{\pm0.38}$ | $70.49_{\pm3.70}$ | $65.41_{\pm3.84}$ |
| FedRep | $66.87_{\pm4.91}$ | $69.07_{\pm5.42}$ | $92.50_{\pm0.65}$ | $94.19_{\pm0.56}$ | $65.27_{\pm1.86}$ | $65.90_{\pm3.39}$ |
| Ditto | $69.05_{\pm4.41}$ | $64.50_{\pm5.09}$ | $91.86_{\pm0.87}$ | $94.93_{\pm0.32}$ | $65.45_{\pm3.43}$ | $65.61_{\pm4.52}$ |
| FedRod | $63.70_{\pm1.96}$ | $77.20_{\pm4.98}$ | $92.64_{\pm0.58}$ | $95.26_{\pm0.31}$ | $73.27_{\pm3.35}$ | $64.88_{\pm4.03}$ |
| Memo(P) | - | - | $92.94_{\pm0.65}$ | $94.48_{\pm0.32}$ | $64.88_{\pm3.13}$ | $62.24_{\pm3.97}$ |
| T3A | $69.80_{\pm1.60}$ | $69.10_{\pm1.50}$ | $91.93_{\pm0.50}$ | $94.20_{\pm0.34}$ | $67.24_{\pm2.01}$ | $65.61_{\pm4.52}$ |
| FedTHE | $70.30_{\pm5.83}$ | $74.97_{\pm3.90}$ | $91.77_{\pm0.85}$ | $94.53_{\pm0.32}$ | $71.80_{\pm3.07}$ | $62.02_{\pm4.22}$ |
| Flute | $70.33_{\pm4.31}$ | $67.04_{\pm3.66}$ | $94.16_{\pm0.69}$ | $94.59_{\pm0.38}$ | $71.61_{\pm1.61}$ | $63.46_{\pm1.58}$ |
| FedSR | $73.88_{\pm3.10}$ | $72.08_{\pm4.85}$ | $93.99_{\pm0.79}$ | $94.22_{\pm0.77}$ | $62.99_{\pm2.11}$ | $\mathbf{68.35}_{\pm0.53}$ |
| CFL | $70.82_{\pm5.43}$ | $70.82_{\pm5.43}$ | $93.84_{\pm0.30}$ | $93.84_{\pm0.30}$ | $64.50_{\pm3.17}$ | $65.28_{\pm3.50}$ |
| CFeD | $68.37_{\pm8.22}$ | $68.38_{\pm8.22}$ | $93.22_{\pm3.21}$ | $94.77_{\pm0.92}$ | $65.30_{\pm2.92}$ | $67.18_{\pm2.91}$ |
| **FedEvolve** | $\mathbf{82.52}_{\pm1.94}$ | $\mathbf{83.59}_{\pm5.91}$ | $93.84_{\pm1.62}$ | $\mathbf{96.54}_{\pm1.39}$ | $\mathbf{75.04}_{\pm4.03}$ | $64.06_{\pm3.83}$ |
| **FedEvp** | $\mathbf{74.80}_{\pm1.69}$ | $\mathbf{77.93}_{\pm4.20}$ | $\mathbf{94.50}_{\pm0.28}$ | $93.91_{\pm2.19}$ | $\mathbf{73.46}_{\pm0.90}$ | $68.24_{\pm1.08}$ |

(a) Dir → ∞      (b) Dir = 1.0      (c) Dir = 0.1

Figure 3: Performance comparison of various methods across different rotation angles on RMNIST for distinct distributions.

with varying degrees of local heterogeneity (Dir = ∞, 1.0, 0.1). Each sub-figure shows how performance changes as the extent of distribution shift changes from no distribution shift (0° incremental angle) to high distribution shift (25° incremental angle):

- In the absence of significant distribution shifts (e.g. rotation incremental angle 0°, 3°, or 5°), Figure 3a shows that, when there is no client heterogeneity, our methods have similar performance as the traditional FL methods. The learning task reduces to the standard FL task, and the classical FL methods maintain competitive performance. As clients get more heterogeneous, Figures 3b and 3c show that all methods experience the accuracy drop and the performance on the server for *FedEvolve* is marginally inferior to that of *FedAvg*, while *FedEvp* with personalization still shows the robustness under heterogeneous clients. This is further verified when Dir = 0.1. We also observe that *FedEvolve* is still robust when client heterogeneity is large. The decline in performance due to heterogeneity mainly comes from the error of the classifier, and *FedEvolve* avoids this by using representation distance to make predictions rather than relying on the classifier.

- When the rotation increments increase, *FedAvg* experiences a significant performance drop (e.g., nearly 12% decrease when the incremental angle increases for 5 degrees, see Figure 3a). Such impacts are more significant than the performance drop caused by client heterogeneity, indicating the challenge of evolving shifts. However, our methods are still robust against such shifts and significantly better than baselines.

When both strong local heterogeneity and distribution shifts are present (Figure 3c), both the baselines and ours experience a performance drop while ours exhibit a relatively slower decline. The better performance on the clients compared to the server for *FedEvp* further validates the effectiveness of the personalization mechanism of *FedEvp*.

**Comparison with Baselines.** We conduct extensive experiments on five datasets with different levels of client heterogeneity. Table 1 and 2 and the results of Circle data in Table 3 compare different methods in scenarios with strong evolving patterns. We observe that both *FedEvolve* and *FedEvp* outperform the baseline methods. In particular, *FedEvolve* attains the highest accuracy (84.75%, 83.58%, and 84.25% on RMNIST, REMNIST, and Circle respectively), demonstrating its capability to learn from the evolving pattern and effectively address the distribution shifts. This advantage also shows on other datasets (Portraits and Caltran) in Table 3 with less obvious evolving patterns.

For PFL or TTA baselines tuned on local source domains, without client heterogeneity (Dir $\to \infty$), the performance may deteriorate compared to classical FL such as *FedAvg*. Specifically, methods such as *FedAvgFT*, *APFL*, and *FedRep* may experience a drop in client performance compared to the server on certain datasets. These methods originally designed to tackle client heterogeneity without learning evolving patterns suffer performance degradation; this further highlights the importance of considering evolving distribution shifts in FL systems. Nonetheless, when clients are heterogeneous (Dir is 1.0 or 0.1 in Table 1 and 2), their personalization or test-time adaptation can still be beneficial.

General domain generalization methods like *FedSR* and continual FL methods tend to achieve better results than other baselines, indicating their capability to mitigate the influence of evolving distribution shifts. But the gap between their performance and that of ours still emphasizes the need for a specific design to solve the problem.

Among all methods, our proposed *FedEvolve* and *FedEvp* show the best performance and are robust to both client heterogeneity and evolving shifts. *FedEvp* achieves comparable performance with *FedEvolve* but only uses half numbers of parameters as *FedEvolve*. Specifically, when Dir = 0.1, *FedEvolve* achieves accuracy of **83.86%** and **87.67%** on RMNIST and REMNIST, while *FedEvp* achieves similar accuracy of **83.15%** and **87.01%**. Thus, a careful design of personalization can prevent the unintended consequence of performance degradation.

**Impact of Straggler.** Stragglers in FL systems introduce heterogeneity at the system level; therefore, we also study how our methods could be resilient to the straggler problem. We report the results in Table 5 when stragglers are present during the training phase. The results show our methods are not significantly affected by stragglers. In this experiment, the straggler ratio represents the probability that a client will train fewer local iterations than the specified number $\tau$. For stragglers, the actual number of local iterations is randomly selected, ranging from 1 to $\tau$.

Table 5: Performance under different straggler ratio.

| R-MNIST(Dir=1.0) | 0 | 0.1 | 0.3 | 0.5 | 0.7 | 0.9 |
|---|---|---|---|---|---|---|
| FedEvolve | $79.93_{\pm 1.00}$ | $78.56_{\pm 4.17}$ | $77.50_{\pm 4.87}$ | $77.42_{\pm 3.45}$ | $75.88_{\pm 2.58}$ | $71.87_{\pm 0.98}$ |
| FedEvp | $77.91_{\pm 1.80}$ | $77.79_{\pm 1.69}$ | $77.11_{\pm 0.83}$ | $77.00_{\pm 1.14}$ | $76.91_{\pm 0.84}$ | $76.60_{\pm 1.54}$ |

**Overhead Comparison.** Table 6 compares transmission overhead. We use an CNN as an example to report the number of parameters and server-client transmission time in the MPI environment. Although *FedEvolve* has the higher transmission overhead, its cost-efficient version *FedEvp* has comparable overhead as the baselines.

Table 6: The number of model parameters and transmission time.

| | FedRod | FedTHE | FedSR | FedEvolve(Ours) | FedEvp (Ours) | Others |
|---|---|---|---|---|---|---|
| Parameters | 382106 | 382208 | 391937 | 741120 | 379392 | 379392 |
| Time/ms | $21.38 \pm 0.45$ | $21.95 \pm 1.23$ | $21.62 \pm 0.87$ | $46.32 \pm 0.78$ | $21.30 \pm 0.86$ | $21.26 \pm 1.11$ |

Table 7: Ablation for *FedEvp* (Dir=0.1). We compare the average accuracy on clients for *FedEvp* with three versions: one without any personalization, another that personalizes only the classifier, and a third that personalizes all parameters.

| Method | MNIST Acc | EMNIST Acc |
|---|---|---|
| FedEvp | $83.15_{\pm 0.49}$ | $87.01_{\pm 0.22}$ |
| FedEvp w/o personalization | $63.59_{\pm 2.38}$ | $57.67_{\pm 1.64}$ |
| FedEvp personalize C | $79.21_{\pm 2.29}$ | $86.59_{\pm 0.35}$ |
| FedEvp personalize all | $73.06_{\pm 1.07}$ | $82.78_{\pm 0.54}$ |

**Ablation study.** We also study the influence of personalization mechanisms of *FedEvp* on the performance in Table 7. The results show personalizing part of the feature extractor and classifier can achieve the best results. We also notice that personalizing the classifier brings the most significant improvement which means the classifier is most sensitive to the client heterogeneity with evolving distribution shifts.

We further investigate the impact of varying the number of source domains on the prediction performance for the target domain. Specifically, we conduct experiments under the same settings as shown in Table 8, while controlling for the number of domains. Our methods are compared to FedAvg using reduced numbers of domains: 7 domains (rotation starting at 75° and increasing to 165°), 10 domains (rotation starting at 30° and increasing to 165°), and 12 domains (rotation starting at 0° and increasing to 165°).

Table 8: Performance under different number of domains.

| R-MNIST(Dir$\to \infty$) | 7 | 10 | 12 |
|---|---|---|---|
| FedAvg | $74.92 \pm 1.08$ | $70.07 \pm 1.48$ | $65.92 \pm 1.01$ |
| FedEvolve | $78.24 \pm 1.18$ | $82.44 \pm 1.40$ | $84.57 \pm 2.45$ |
| FedEvp | $80.20 \pm 2.09$ | $74.17 \pm 1.15$ | $75.99 \pm 0.31$ |
| R-MNIST(Dir=1.0) | 7 | 10 | 12 |
| FedAvg | $71.44 \pm 1.21$ | $65.40 \pm 0.58$ | $62.35 \pm 0.97$ |
| FedEvolve | $75.07 \pm 2.42$ | $80.81 \pm 2.33$ | $79.93 \pm 1.00$ |
| FedEvp | $80.19 \pm 1.26$ | $78.99 \pm 0.82$ | $77.91 \pm 1.80$ |
| R-MNIST(Dir=0.1) | 7 | 10 | 12 |
| FedAvg | $61.18 \pm 0.91$ | $54.83 \pm 1.24$ | $51.68 \pm 0.73$ |
| FedEvolve | $78.56 \pm 1.69$ | $83.44 \pm 2.25$ | $83.86 \pm 1.81$ |
| FedEvp | $78.16 \pm 4.26$ | $83.92 \pm 1.59$ | $82.12 \pm 1.84$ |

As the number of domains increases, FedAvg shows significant performance degradation across all heterogeneity settings. This indicates standard methods' vulnerability to evolving distributional shifts. Both FedEvolve and FedEvp display robustness against increasing domain numbers, maintaining or improving performance. In particular, by incorporating more source domains, FedEvolve is able to fully learn the transition of two consecutive domains. However, FedEvp remains less sensitive to domain transitions, performing consistently well across different settings. The robustness of our methods contrasts sharply with the performance drop observed in FedAvg, highlighting the importance of handling distribution variability in federated learning.

**Discussion.** The empirical evidence suggests that conventional FL algorithms cannot simultaneously handle the evolving distributional shifts and clients heterogeneity. In addition, evolving distributional shifts could be viewed as a specific form of data heterogeneity affecting client devices. Present personalization strategies, designed for data heterogeneity, fail in adapting models to unseen distributions. Simply tuning clients on known domains without considering shifts between training data and test data, these methods may inadvertently increase the model's bias towards training data resulting in performance that is sometimes inferior to that of non-personalized algorithms. While continual FL frameworks take account of dynamic distributional shifts during training, they primarily concentrate on preventing catastrophic forgetting of prior tasks or domains rather than adapting to new, unseen ones. This focus makes them inadequate for managing evolving distributional shifts effectively. However, when the distribution of a target domain is

predictable based on existing data, our methods explicitly leverage and learn the pattern of distribution transitions, enabling the extrapolation of the model to the target domain. Therefore, our methods mitigate the performance drop and achieve the best results.

## 6 Conclusions

This paper studies FL under evolving distribution shifts. We explored the impacts of evolving shifts and client heterogeneity on FL systems and proposed two algorithms: *FedEvolve* that precisely captures the evolving patterns of two consecutive domains, and *FedEvp* that learns a domain-invariant representation for all domains with the aid of personalization. Extensive experiments show both algorithms have superior performance compared to SOTA methods.

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

# A   Algorithm

We present the pseudo-code for FedEvolve and FedEvp in Alg.1 and Alg.2. We randomly sample a subset of data from the dataset to train the model for each update instead of the whole dataset.

---
**Algorithm 1** FedEvolve
---

**Require:** Number of clients $K$; client participation ratio $r$; step size $\eta$; the number of local training updates $\tau$; communication rounds $T$; the number of source domains $M$; initial global parameter $\phi$ and global parameter $\psi$; local datasets $\mathcal{D}_m^k$ and their known classes $\mathcal{Y}_{\mathcal{D}_m^k}$ for $m \in \{1, \ldots, M\}, k \in \{1, \ldots, K\}$.

1: **for** $t \in \{1, \ldots, T\}$ **do**
2:     server samples $rK$ clients as $\mathcal{I}_t$ from all clients
3:     server sends $\phi$, $\psi$ to $\mathcal{I}_t$
4:     **for** each client $k \in \mathcal{I}_t$ in parallel **do**
5:         client $k$ initialize $\widetilde{\phi}_k := \phi$, $\widetilde{\psi}_k := \psi$
6:         **for** $\tau$ local training iterations **do**
7:             **for** $m \in \{1, \ldots, M-1\}$ **do**
8:                 $\mathcal{A} \leftarrow RandomSample(\mathcal{D}_m^k)$
9:                 $\mathcal{B} \leftarrow RandomSample(\mathcal{D}_{m+1}^k)$
10:                **for** $y \in \mathcal{Y}_{\mathcal{D}_m^k}$ **do**
11:                    $\mathcal{A}_y \leftarrow \{(x_i, y_i) \in \mathcal{A} | y_i = y\}$
12:                    $C_{m,y}^k = \frac{1}{|\mathcal{A}_y|} \sum_{(x_i, y_i) \in \mathcal{A}_y} f_{\widetilde{\phi}_k}(x_i)$
13:                **end for**
14:                $\ell = 0$
15:                **for** $(x, y) \in \mathcal{B}$ **do**

16:                $$\ell = \ell + \frac{1}{|\mathcal{B}|}\Big[\log \frac{\exp\Big(-d\Big(f_{\widetilde{\psi}_k}(x), C_{m,y}^k\Big)\Big)}{\sum_{y' \in \mathcal{Y}_{\mathcal{D}_m^k}} \exp\Big(-d\Big(f_{\widetilde{\psi}_k}(x), C_{m,y'}^k\Big)\Big)\Big)}\Big]$$

17:                **end for**
18:                $\widetilde{\phi}_k, \widetilde{\psi}_k = GradientDescent(\ell; \widetilde{\phi}_k, \widetilde{\psi}_k, \eta)$
19:            **end for**
20:        **end for**
21:        client $k$ sends local parameters $\widetilde{\phi}_k, \widetilde{\psi}_k$ to server
22:    **end for**
23:    $\phi = \frac{1}{|\mathcal{I}_t|} \sum_{k \in \mathcal{I}_t} \widetilde{\phi}_k$
24:    $\psi = \frac{1}{|\mathcal{I}_t|} \sum_{k \in \mathcal{I}_t} \widetilde{\psi}_k$
25: **end for**
26: **Output** $\phi$ and $\psi$

---

---

**Algorithm 2** FedEvp

---

**Require:** Number of clients $K$; client participation ratio $r$; step size $\eta$; the number of local training updates $\tau$; communication rounds $T$; the number of source domains $M$; initial global parameter $\phi$ and global parameter $\psi$; local datasets $\mathcal{D}_m^k$ and their known classes $\mathcal{Y}_{\mathcal{D}_m^k}$ for $m \in \{1, \ldots, M\}, k \in \{1, \ldots, K\}$.

1: **for** $t \in \{1, \ldots, T\}$ **do**
2:     server samples $rK$ clients as $\mathcal{I}_t$ from all clients
3:     server sends $\phi, \psi$ to $\mathcal{I}_t$
4:     **for** each client $k \in \mathcal{I}_t$ in parallel **do**
5:       client $k$ initialize $\widetilde{\phi}_k := \phi$, $\widetilde{w}_k := w$
6:       **for** $\tau$ local training iterations **do**
7:         **for** $y \in \mathcal{Y}_{\mathcal{D}_m^k}$ **do**
8:           $C_{0,y}^k = 0$
9:         **end for**
10:        **for** $m \in \{1, \ldots, M\}$ **do**
11:          $\mathcal{A} \leftarrow RandomSample(\mathcal{D}_m^k)$

12:          $\ell_e \leftarrow -\frac{1}{|\mathcal{A}|} \sum\limits_{(x_i, y_i) \in \mathcal{A}} y_i \log \dfrac{\exp\left(g_{\widetilde{w}_k}^y \left(f_{\widetilde{\phi}_k}(x)\right)\right)}{\sum_{y' \in \mathcal{Y}_{\mathcal{D}_m^k}} \exp\left(g_{\widetilde{w}_k}^{y'}\left(f_{\widetilde{\phi}_k}(x)\right)\right)}$

13:          **for** $y \in \mathcal{Y}_{\mathcal{D}_m^k}$ **do**
14:            $\mathcal{A}_y \leftarrow \{(x_i, y_i) \in \mathcal{A} | y_i = y\}$
15:            $C_{m,y}^k = \frac{(m-1)}{m} C_{m-1,y}^k + \frac{1}{m} \frac{1}{|\mathcal{A}_y|} \sum_{(x_i, y_i) \in \mathcal{A}_y} f_{\widetilde{\phi}_k}(x_i)$
16:          **end for**
17:          **if** m$\geq$2 **then**
18:            $\ell_f = 0$
19:            **for** $(x, y) \in \mathcal{A}$ **do**

20:            $\ell_f = \ell_f + \frac{1}{|\mathcal{A}|} \log \dfrac{\exp\left(-d\left(f_{\widetilde{\phi}_k}(x), C_{m,y}^k\right)\right)}{\sum_{y' \in \mathcal{Y}_{\mathcal{D}_m^k}} \exp\left(-d\left(f_{\widetilde{\phi}_k}(x), C_{m,y'}^k\right)\right))}$

21:            **end for**
22:            $\widetilde{\phi}_k, \widetilde{w}_k = GradientDescent(\ell_f + \ell_e; \widetilde{\phi}_k, \widetilde{w}_k, \eta)$
23:          **end if**
24:          **end for**
25:        **end for**
26:       client $k$ sends local parameters $\widetilde{\phi}_k, \widetilde{w}_k$ to server
27:     **end for**
28:     $\phi = \frac{1}{|\mathcal{I}_t|} \sum_{k \in \mathcal{I}_t} \widetilde{\phi}_k$
29:     $w = \frac{1}{|\mathcal{I}_t|} \sum_{k \in \mathcal{I}_t} \widetilde{w}_k$
30: **end for**
31: **Server Output** $\phi, w$
32: **for** each client $k$ **do**
33:     **Client Output** $\widetilde{\phi}_k, \widetilde{w}_k = $ personalize$(\phi, w, \mathcal{D}^k)$
34: **end for**

---

# B   Datasets

### B.1   Rotated MNIST (Ghifary et al., 2015) and Rotated EMNIST (Ghifary et al., 2015)

For Rotated MNIST (RMNIST), We generate 12 domains by applying the rotations with angles of $\theta = \{0°, 15°, ..., 165°\}$ on each domain respectively. For Rotated EMNIST (REMNIST), we generate 12 domains by applying the rotations with angles of $\theta = \{0°, 8°, ..., 88°\}$ on each domain respectively.

### B.2   Circle (Pesaranghader & Viktor, 2016)

We follow (Pesaranghader & Viktor, 2016) to generate this dataset. In this synthetic data set, we have 30 Gaussian distributions centered on a half circle with standard deviation 0.6, and the radius $r$ is set to 10. Each data point has two attributes, and the number of classes is 2. The decision boundary is $(x - x_0)^2 + (y - y_0)^2 \leq r^2$, where $(x_0, y_0)$ are the coordinates of the circle's center (we set it as $(0,0)$).

### B.3   Portraits (Ginosar et al., 2015)

The portraits dataset contains human face images from yearbooks spanning from 1905 to 2013. We partition the data into nine domains by segmenting the dataset into 12-year intervals. All images are resized into 32×32 without any augmentation.

### B.4   Caltran(Hoffman et al., 2014)

This real surveillance dataset comprises images captured by a fixed traffic camera deployed in an intersection. The images in this dataset come with time attributes. We categorize the images into 12 distinct domains based on their capture time throughout the day. Specifically, each domain represents a 2-hour interval. As such, a 24-hour day is evenly divided into these 12 domains. We resize images in Caltran to 224×224.

## C   Implementation

All experiments are conducted on a server equipped with multiple NVIDIA A5000 GPUs, two AMD EPYC 7313 CPUs, and 256GB memory. The code is implemented with Python 3.8 and PyTorch 1.13.0 on Ubuntu 20.04 based on the implementation in Jiang & Lin (2023). To evaluate our methods, we consider classification tasks using various network architectures and report average accuracy over three different random seeds. Due to the constraints of our computing resources, our experiments involve between 10 to 20 clients and are conducted over 50 communication rounds, the personalization epoch is 1 for PFL methods including FedEvp.

- RMNIST: For the Rotated MNIST dataset, we employ a CNN with four convolutional layers, each equipped with a 3x3 kernel. Group Normalization is applied post-convolution for stabilization using groups of 8 channels. Followed by convolutional layers, there are two linear layers with a hidden dimension of 64. The four convolutional layers and the first linear layer form the representation functions ($f_\phi$ and $f_\psi$ in FedEvolve, $f_\phi$ in FedEvp) with the final linear layer serving as the classifier ($f_w$ in FedEvp). We employ an SGD optimizer with a weight decay of 5e-4 and conduct local training for 10 epochs per communication round.

- **REMNIST**: For the Rotated EMNIST dataset, we employ the same CNN as the one in RMNIST. We use Adam optimizer with weight decay of 1e-4 and run local training for 10 epochs per communication round.

- **Circle:** For the Circle dataset, we utilize a five-layer Multi-Layer Perceptron (MLP). This includes three dense layers (2x256, 256x256, 256x256) for feature representation ($f_\phi$ and $f_\psi$ in FedEvolve, $f_\phi$ in FedEvp), linked by ReLU activations, and two subsequent linear layers (256x64, 64x2) that function as the classifier ($f_w$ in FedEvp). Given data constraints, we utilize 10 clients and train for 5 epochs using Adam with a weight decay of $5 \times 10^{-4}$.

- **Portraits**: For this dataset, images are resized to 32x32 and processed using a WideResNet architecture. The convolution layers along with the average pooling layer act as the representation function. A linear layer is designated as the classifier after representation. The model is trained among 20 clients over 5 epochs using an Adam optimizer with a weight decay of 5e-4.

- **Caltran:** We deploy ResNet18 for the Caltran dataset, with the last linear layer used as the classifier. The representation function comprises four residual convolution blocks and an average pooling layer. With pre-trained weights, we tune the model using an SGD optimizer with a weight decay of 5e-4. Given data limitations, training involves 10 clients over 5 epochs.

Networks for each dataset are presented in Table 9

| Dataset | Input Dimension | Number of Classes | Network |
|---|---|---|---|
| RMNIST | $28 \times 28$ | 10 | CNN |
| REMNIST | $28 \times 28$ | 26 | CNN |
| Circle | 2 | 2 | MLP |
| Portraits | $32 \times 32$ | 2 | WideResNet |
| Caltran | $3 \times 224 \times 224$ | 2 | ResNet18 |

Table 9: Networks for datasets

For each dataset, we search the learning rate for each algorithm to find the best results. The training detail is given in Table 10.

| Dataset | Num of Clients | Batch Size | Learning Rate Range |
|---|---|---|---|
| RMNIST | 20 | 32 | 1e-3, 1e-2, 1e-1 |
| REMNIST | 20 | 96 | 1e-3, 5e-3, 1e-2, 5e-2, 1e-1 |
| Circle | 10 | 32 | 1e-6, 5e-6, 1e-5, 5e-5, 1e-4 |
| Portraits | 20 | 32 | 1e-3, 5e-3, 1e-2 |
| Caltran | 10 | 32 | 1e-5, 5e-5, 1e-4, 5e-4 |

Table 10: Training Details for datasets

We use the same search strategy for hyperparameters to tune the models.

- For GMA(Tenison et al., 2022), we set the masking threshold as 0.1, searching from $\{0.1, 0.2, 0.3, ..., 1.0\}$

- For FedRep(Collins et al., 2021), FedRod(Chen & Chao, 2022), and FedTHE(Jiang & Lin, 2023), the last fc layer of the model is used as the head.

- For Ditto(Li et al., 2021a), the regularization factor $\lambda$ is set to 0.1.

- For MEMO,(Zhang et al., 2021b) we use 32 augmentations and 3 optimization steps.

- For T3A(Iwasawa & Matsuo, 2021), $M = 50$ is used in our experiments.

- For FedSR(Nguyen et al., 2022), we follow the same setting in their paper: $\alpha^{L2R} = 0.01$ and $\alpha^{CMI} = 0.001$.

## D   Supplementary Results

We compared the P-values of our proposed methods, FedEvolve and FedEvp, with various baseline federated learning algorithms in Table 11. The p-values from our t-test statistical analysis indicated that our methods significantly outperform the baseline methods.

Table 11: P-values comparing FedEvolve and FedEvp with baseline methods on rotated MNIST.

| | FedAvg | GMA | Memo(G) | FedAvgFT | APFL | FedRep | Ditto |
|---|---|---|---|---|---|---|---|
| FedEvolve | $5.17 \times 10^{-4}$ | $4.42 \times 10^{-4}$ | $7.69 \times 10^{-4}$ | $1.44 \times 10^{-3}$ | $2.26 \times 10^{-4}$ | $9.41 \times 10^{-4}$ | $7.27 \times 10^{-4}$ |
| FedEvp | $7.33 \times 10^{-3}$ | $6.76 \times 10^{-3}$ | $9.82 \times 10^{-3}$ | $2.42 \times 10^{-3}$ | $6.20 \times 10^{-6}$ | $4.60 \times 10^{-4}$ | $2.16 \times 10^{-5}$ |
| | FedRod | Memo(P) | T3A | FedTHE | FedSR | CFL | CFeD |
| FedEvolve | $6.21 \times 10^{-4}$ | $1.04 \times 10^{-3}$ | $8.09 \times 10^{-4}$ | $8.92 \times 10^{-4}$ | $6.27 \times 10^{-4}$ | $2.46 \times 10^{-4}$ | $1.41 \times 10^{-3}$ |
| FedEvp | $2.71 \times 10^{-3}$ | $1.20 \times 10^{-3}$ | $7.71 \times 10^{-4}$ | $2.27 \times 10^{-4}$ | $2.77 \times 10^{-2}$ | $4.04 \times 10^{-3}$ | $4.40 \times 10^{-2}$ |

Table 12: Average accuracy over three runs of experiments on rotated MNIST with different rotation degrees for the target domain.

| | Dir$\to\infty$ | | Dir=1.0 | | Dir=0.1 | |
|---|---|---|---|---|---|---|
| Method | Client | Server | Client | Server | Client | Server |
| FedAvg (110°) | $80.96_{\pm1.62}$ | $80.92_{\pm0.23}$ | $79.41_{\pm0.30}$ | $81.63_{\pm1.71}$ | $70.16_{\pm0.48}$ | $71.93_{\pm2.06}$ |
| FedAvg (120°) | $66.39_{\pm0.92}$ | $66.61_{\pm1.17}$ | $64.30_{\pm1.36}$ | $65.08_{\pm1.89}$ | $54.74_{\pm1.40}$ | $53.62_{\pm1.01}$ |
| FedAvg (130°) | $50.53_{\pm0.88}$ | $51.04_{\pm2.72}$ | $48.64_{\pm1.28}$ | $48.41_{\pm1.16}$ | $40.87_{\pm1.44}$ | $40.41_{\pm0.91}$ |
| FedEvolve (110°) | $86.66_{\pm0.66}$ | $86.62_{\pm1.60}$ | $85.57_{\pm1.35}$ | $83.11_{\pm2.31}$ | $86.92_{\pm1.72}$ | $74.67_{\pm2.29}$ |
| FedEvolve (120°) | $78.09_{\pm0.88}$ | $77.80_{\pm0.82}$ | $74.85_{\pm2.86}$ | $72.81_{\pm5.46}$ | $78.43_{\pm4.92}$ | $61.57_{\pm7.44}$ |
| FedEvolve (130°) | $65.13_{\pm1.79}$ | $64.64_{\pm1.85}$ | $62.88_{\pm2.23}$ | $60.47_{\pm3.91}$ | $69.77_{\pm4.30}$ | $50.82_{\pm5.58}$ |
| FedEvp (110°) | $84.68_{\pm1.51}$ | $86.07_{\pm1.38}$ | $85.84_{\pm1.33}$ | $83.33_{\pm3.50}$ | $84.99_{\pm2.32}$ | $69.41_{\pm2.47}$ |
| FedEvp (120°) | $72.91_{\pm0.65}$ | $75.66_{\pm0.82}$ | $74.93_{\pm2.54}$ | $71.82_{\pm2.83}$ | $79.48_{\pm1.89}$ | $62.28_{\pm3.88}$ |
| FedEvp (130°) | $61.67_{\pm0.31}$ | $64.64_{\pm0.16}$ | $65.79_{\pm2.36}$ | $62.99_{\pm2.11}$ | $72.11_{\pm3.01}$ | $53.84_{\pm4.05}$ |

## D.1  Impact of Changing Pattern

In previous experiments, we primarily focus on invariant changing patterns in image rotation experiments. Here we examine if our methods are robust against an unexpected pattern. In this experiment, we test the robustness of our methods against an unexpected pattern. Specifically, we simulate an unexpected domain by rotating images from the target domain by an additional 10° and 20°. To prevent confusion between numerals like 6 and 9 when rotated by 180°, we set the incremental rotation degree as 10°. Therefore, the images experience a 120° rotation or a 130° rotation instead of the expected 110°. This experiment aims to evaluate whether our methods can handle deviations from anticipated patterns.

As shown in Table 12, all methods exhibit a significant performance drop when the test data distribution changes substantially, however, our methods still outperform the baseline and the drop is less than the baseline. Notably, FedEvp demonstrates superior performance compared to FedEvolve when clients are heterogeneous. This difference arises because FedEvolve explicitly learns the distribution transition between consecutive domains, while FedEvp learns evolving-domain-invariant features. Consequently, when the distribution transition deviates from the learned pattern, the performance of FedEvolve is adversely affected, whereas FedEvp remains less influenced by the change.

