# OpenReview forum: "Federated Learning under Evolving Distribution Shifts"
_TMLR — Rejected by TMLR_

### Review · Reviewer_UCDx · 2024-05-29

**Summary Of Contributions:**

This paper aims to tackle the challenge where the client data distributions changeover time, and the dynamics, i.e., the evolving patter and the distribution shift between the training and testing data in the federated learning framework. Two new approaches named FedEvolve and FedEvp are proposed to capture the evolving patterns of the clients during training and are test-robust under evolving distribution shifts. Experimental results show the effectiveness of the proposed methods.

**Audience:**

Yes

**Broader Impact Concerns:**

I do not foresee adverse ethical implications of the proposed work.

**Claims And Evidence:**

Yes

**Requested Changes:**

- In the current setting, the authors implicitly consider the invariant changing pattern, suggesting that the evolving patterns remain constant despite the distribution shifts. How would the methods handle a scenario where the changing pattern itself changes? Please discuss this aspect in your revision.

- Is there any real time-series data available to test the performance of your methods?

- Would using more domains to predict the new domain representation enhance performance?

- Could you discuss "Continuously Indexed Domain Adaptation" and explore whether their techniques could be beneficial for your approach?

**Strengths And Weaknesses:**

I have few technical comments. Combining the federated setting with evolving distribution shifts is logical. However, the proposed setting appears to be a straightforward combination of the two without introducing significant challenges, at least as per the methods presented in this paper. While the technical aspects don't particularly stand out to me, it is acceptable to initially formulate this problem and design a model to address this scenario.

---

> ### Author Response · Authors · 2024-07-24
>
> Dear Reviewer UCDx,
>
> Thanks for the comments. We address your concerns point by point as follows.
>
> ## How would the methods handle a scenario where the changing pattern itself changes? Please discuss this aspect in your revision.
>
> Thanks for your suggestion. We consider invariant changing patterns in MNIST-based experiments. However, the patterns in the Portraits and Caltran datasets change more naturally. To further address your concern, in our revised manuscript, we included new experimental results in Appendix D.1 where we evaluate our methods under unexpected domain changes. When the distribution transition significantly deviates from the learned pattern, the performance of FedEvolve is adversely affected because relies on a learned domain shift pattern for prediction, whereas, FedEvp remains less influenced by learning domain-invariant representations.
>
> ## Is there any real time-series data available to test the performance of your methods?
>
> Time-series forecasting is not in the objectives of our proposed methods. However, we already conducted experiments on datasets with time attributes. For example, the *Portraits* dataset consists of images with year labels. We split the dataset based on the time sequence and tested our model on unseen time periods.
>
> ## Would using more domains to predict the new domain representation enhance performance?
>
> We conducted extra experiments on Rotated MNIST to examine the performance under different numbers of domains. The results are presented in Table 8 of the revised manuscript. Using more domains is beneficial for our methods to achieve better performance on the new domain, especially using FedEvolve which directly learns the domain transition.
>
> ## Could you discuss "Continuously Indexed Domain Adaptation" and explore whether their techniques could be beneficial for your approach?
>
> Thanks for sharing the related paper. Continuously Indexed Domain Adaptation is relevant to our work as we both consider a series of domains. The main difference is that CIDA is allowed access to unlabeled data and domain indices in target domains during training while our setting is more close to domain generalization. Therefore, CIDA cannot be directly used in our setting. However, the idea of using discriminators to learn domain-invariant features can be extended to our approach. We have updated our manuscript and discussed them in the related work section.

---

> > ### Comment · Reviewer_UCDx · 2024-07-26
> > **No further quesitons**
> >
> > Thank you for your detailed response. I have no further questions.

---

### Review · Reviewer_HRZK · 2024-06-03

**Summary Of Contributions:**

The paper considers federated learning for non-stationary and heterogeneous client data distributions. The non-stationarity covers domain shifts at predefined time steps. Two algorithms are proposed to deal with the variation in the data distribution. Empirical evaluations with synthetic and image based data show that the proposed algorithms perform better than alternative federated learning algorithm.

**Audience:**

Yes

**Broader Impact Concerns:**

None.

**Claims And Evidence:**

No

**Requested Changes:**

Defining the algorithm in a more precise way with all the necessary details is really a must.

I would also suggest the authors to provide access to the source code and data sets.

**Strengths And Weaknesses:**

Enabling federated learning to work under realistic conditions when the data distributions are not static (and not homogeneous) is certainly important. In many domains, data distributions change gradually and at time steps that are not known a priori, therefore the setting is somewhat limited. Nevertheless, there are domains where data arrives in batches (perhaps from different sources), and the setting discussed can be relevant.

The presentation of the algorithms is rather superficial with too many details left out or specified informally only. This include the imprecise definition of the mapping functions, the description of the training algorithm, and even the pseudocode in the Appendix still leaves functions such as Update in the air.

Source code is not provided, and given the superficial description of the algorithms, the experiments would be rather difficult to replicate. Given the profile of journal, this is a considerable weakness for a submission with largely empirical contribution.

The number of baselines is reasonable. I would have been tempted to include centralized algorithms as well.

---

> ### Author Response · Authors · 2024-07-24
>
> Dear Reviewer HRZK,
>
> Thanks for the comments. We address your concerns point by point as follows.
>
> ## Defining the algorithm in a more precise way with all the necessary details is really a must.
>
> We have updated the manuscript to define our algorithm in a more precise way.
>
> ## I would also suggest the authors to provide access to the source code and data sets.
>
> Thanks for your suggestion. We have uploaded our core code at https://anonymous.4open.science/r/3D68. The full version will be released on GitHub after acceptance.
>
> ## The number of baselines is reasonable. I would have been tempted to include centralized algorithms as well.
>
> Some of our baselines are centralized algorithms and adapted to FL settings, such as MEMO and T3A.

---

### Review · Reviewer_rczs · 2024-07-11

**Summary Of Contributions:**

The paper addresses a problem of heterogeneity of clients' data distribution that is non-trivially evolving in time. To tackle the challenges, the paper proposes two algorithms -- FedEvolve and FedEvp. The FedEvolve learns two different representation mappings of previous and current time step's data by the expense of computational and storage cost of maintaining two representation spaces. The FedEvp maintains only a single space for efficiency. In the empirical evaluations, the proposed methods outperforms the prior arts by noticeable margins.

**Audience:**

Yes

**Claims And Evidence:**

Yes

**Requested Changes:**

- Add analysis why this method should be the best way so far to address this tasks
- Compare with more recent methods

**Strengths And Weaknesses:**

**Strengths**
- Application of time evolving distribution domain adaptation to federated learning context.
- Empirical results seems good

**Weaknesses**
- Technical contribution is weak. The methods are quite common, similar to conventional large margin embedding (Weinberger and Chapelle, Large Margin Taxonomy Embedding with an Application to Document Categorization, NeurIPS 2008)
- Compared methods are one year old. In the rapidly changing field of ML, there are a number of relevant methods to be compared. To name a few,
	- Abdelmoniem et al, Resource-Efficient Federated Learning, ICML 2023 workshop
	- Kim et al., Clustered Federated Learning via Gradient-based Partitioning, ICML 2024
	- Yang et la., SimFBO: Towards Simple, Flexible and Communication-efficient Federated Bilevel Learning, NeurIPS 2023
- Contents of the paper is quite slim as if it is a conference submission. There is no in-depth analysis or deep insight about a problem typically shown in a journal paper.

---

> ### Author Response · Authors · 2024-07-24
>
> Dear Reviewer rczs,
>
> Thanks for the comments. We address your concerns point by point as follows.
>
> ## Relationship to large margin embedding
>
> We want to emphasize our study is focusing on evolving distributional shifts in Federated Learning. We leverage prototype-based continually aligning methods to learn the evolving pattern. Large-margin embedding is a widely study subject for adjusting sample embedding distance using techniques like contrastive loss or triplet loss. However, prototype learning aims to learn a set of prototypes (i.e., centroids) that represent the center of each class in the latent space. In this study, we propose a new prototype learning method that dynamically updates the prototypes to adapt to different domains and address distributional shifts effectively.
>
> ## New baselines
>
> We apologize for the earlier mistake in denoting the publication date of the FedTHE paper. It was published in ICLR 2023, not 2022 as previously stated. At the time we evaluated our methods, FedTHE was the state-of-the-art personalization method.
> Thanks for suggesting the three papers. *Clustered Federated Learning via Gradient-based Partitioning* is partly related to our study topic. However, its code was unavailable when we were writing this response. *Resource-Efficient Federated Learning* studies the selection of clients and updates from straggling participants, while our work primarily focuses on the distributional shift problem and does not consider those challenges. *SimFBO: Towards Simple, Flexible and Communication-efficient Federated Bilevel Learning* proposes a new framework to speed up convergence for Federated Bilevel Learning while Federated Bilevel Learning is not relevant to our study.
>
> We are sorry that we are unable to directly compare our methods with these methods. To address your concern, we have added *Flute* [1] as a new, most recent baseline.
>
> [1] Liu, Renpu, Cong Shen, and Jing Yang. "Federated Representation Learning in the Under-Parameterized Regime."  *Forty-first International Conference on Machine Learning* , 2024
>
> ## Requested Changes
>
> We have provided more analysis and included a new baseline in the revision.

---

### Comment · Action_Editor_Joz4 · 2024-07-12
**Rolling discussion**

Dear authors,

There are three reviews included. Could you try to provide rebuttals and do rolling discussions with reviewers within two weeks?

Best wishes,
AC

---

### Decision · Action_Editor_Joz4 · 2024-09-03

**Recommendation:** Reject

**Comment:**

After the rebuttal, Reviewer HRZK and Reviewer rczs still recommend rejection due to the paper lacking in-depth analysis. Existing heuristic methods may not be of interest to TMLR audiences. The action editor has read the rebuttal materials and the paper carefully and found that:
- The research problem is not well-supported.
- The motivation for leveraging prototypical learning rather than other domain adoption methods in their method is unclear.
- The explanation of why the method works well and under what conditions it might fail is not clear.
Based on its current form, the action editor recommends rejection but encourages the authors to resubmit after making significant revisions.

**Audience:**

The paper in its current form may not sufficiently capture the interest of TMLR’s audience in Federated Learning. Addressing the concerns mentioned in Claims And Evidence could make it more appealing.

**Claims And Evidence:**

This paper combines a domain adaptation technique for Federated Learning. However, some major claims are not well supported.
1. The authors claim that no existing work combines domain adaptation for Federated Learning. However, there are papers that consider heterogeneity shift and use domain adaptation methods for it, such as:
- Diurnal or Nocturnal? Federated Learning from Periodically Shifting Distributions
- Is Heterogeneity Notorious? Taming Heterogeneity to Handle Test-Time Shift in Federated Learning
- Spatio-temporal Heterogeneous Federated Learning for Time Series Classification with Multi-view Orthogonal Training
2. The motivation for using prototypical learning in their method is unclear. There are many domain adaptation methods available, why the authors chose to align their representation prototypes without comparing to other advanced methods is not explained. They could consider tweaking different domain adaptation methods in the FL setting to demonstrate that their chosen method outperforms others.
3. All reviewers appreciate the empirical performance. However, the paper lacks a detailed explanation of why their method can improve performance and under what conditions it might fail.

Given its current state, we recommend a major revision. However, since our system does not offer this option explicitly, we encourage the authors to resubmit their work after the revisions (Resubmission Of Major Revision).

**Resubmission Of Major Revision:**

The authors may consider submitting a major revision at a later time.